# Evaluation of Long-Term Performance of Six PM_2.5_ Sensor Types

**DOI:** 10.3390/s25041265

**Published:** 2025-02-19

**Authors:** Karoline K. Barkjohn, Robert Yaga, Brittany Thomas, William Schoppman, Kenneth S. Docherty, Andrea L. Clements

**Affiliations:** 1US Environmental Protection Agency Office of Research and Development, Research Triangle Park, Durham, NC 27711, USA; docherty.kenneth@epa.gov; 2Amentum, Research Triangle Park, Durham, NC 27711, USA; yaga.robert@epa.gov (R.Y.); thomas.brittany@epa.gov (B.T.); schoppman.william@epa.gov (W.S.)

**Keywords:** air sensor, PM_2.5_, air quality, air monitoring, data, quality assurance, colocation

## Abstract

From July 2019 to January 2021, six models of PM_2.5_ air sensors were operated at seven air quality monitoring sites across the U.S. in Arizona, Colorado, Delaware, Georgia, North Carolina, Oklahoma, and Wisconsin. Common PM sensor data issues were identified, including repeat zero measurements, false high outliers, baseline shift, varied relationships between the sensor and monitor, and relative humidity (RH) influences. While these issues are often easy to identify during colocation, they are more challenging to identify or correct during deployment since it is hard to differentiate between real pollution events and sensor malfunctions. Air sensors may exhibit wildly different performances even if they have the same or similar internal components. Commonly used RH corrections may still have variable bias by hour of the day and seasonally. Most sensors show promise in achieving the U.S. Environmental Protection Agency (EPA) performance targets, and the findings here can be used to improve their performance and reliability further. This evaluation generated a robust dataset of colocated air sensor and monitor data, and by making it publicly available along with the results presented in this paper, we hope the dataset will be an asset to the air sensor community in understanding sensor performance and validating new methods.

## 1. Introduction

Air sensors are increasingly used to measure particulate matter (PM) around the United States and the world. Air sensors, sometimes called “low-cost sensors”, can cost an order of magnitude less than air monitors and require fewer resources to operate and maintain. However, many air sensors have limitations in accuracy and precision that may make it challenging to deliver credible data [1,2]. In addition, there can be a lack of information from manufacturers on factory calibrations and other design features that can impact sensor performance. Sensor data must be carefully examined to identify problems [3] and understand limitations [4]. Particulate matter air sensors typically measure particles using light scattering either from a cloud of particles (i.e., nephelometric) or from single particles (i.e., optical particle counter (OPC)) [5]. Air sensors can fill in spatial gaps between regulatory monitors once their limitations are understood and improved where possible.

Air sensors are typically operated alongside federal reference method (FRM), federal equivalent method (FEM) monitors, or other research-grade monitors (i.e., colocation) to better understand their performance [6,7]. Colocating air sensors with FRMs or FEMs, or other air monitors enables users to understand better the performance of the sensors (e.g., accuracy, precision, bias, and drift) relative to the monitors and to understand the influence of other environmental factors on performance [8,9,10,11]. Colocated data (i.e., within 20 m horizontal of FRM or FEM monitors [10]) are scarce, especially for time periods greater than a few months. Colocation data are valuable for air sensor evaluations, quantifying precision between sensors, refining air sensor quality control algorithms, determining air sensor correction algorithms, estimating the uncertainty of data, and other validation activities. To date, many multi-sensor evaluations occur at a single site [12,13]. Single-site and short-term evaluations have similar limitations because of the limited range of environmental conditions and particle properties experienced. Sensor evaluations considering multiple sites often consider a single sensor manufacturer or a single model of air sensor [14,15,16,17,18,19], providing less generalizable results. Recent work has identified nearby air sensor air monitor pairs and assumed they are close enough to make strong comparisons [14,18]. While this may often be the case, localized sources and real differences in pollutant concentrations can bias the findings. There is a need for more multi-site, multi-sensor, longer-term true colocation data to understand air sensor performance better, identify failure patterns, build effective corrections, and make sensors more robust, enabling a variety of air monitoring applications.

This evaluation across multiple U.S. states expands our understanding of PM_2.5_ air sensor performance by operating multiple sensor types across the United States, thus exposing them to a large range of environmental conditions and aerosol composition for more than a year. Our objectives with this study are (1) to identify common failure modes of PM sensors, (2) to briefly explore the influence of RH on various PM_2.5_ air sensors, (3) to compare bias across geographically diverse sites at different times of the year, and (4) to provide a publicly accessible dataset for validation of data quality assurance and correction methods.

## 2. Materials and Methods

### 2.1. Study Design Overview

Six sensor types were colocated across seven air monitoring sites. Nine sensors of each type were first colocated in Research Triangle Park, North Carolina (NC) (prior to July 2019), and then typically, one of each sensor type was deployed to the other six air monitoring sites while three remained in NC. Concurrent colocations across the seven sites ran between 22 July 2019 and 1 January 2021; there were technical difficulties in making the Arisense (ARS) (Aerodyne, Billerica, MA, USA) sensors operational, so those devices were sent out later in February 2020, and sixteen ARS sensors were evaluated in total with the additional sensors deployed in NC. In total, 58 sensors were evaluated. At the end of this study, all sensors were brought back to NC for a final 30-day colocation.

### 2.2. Sensors Selected

Six sensor models were selected based on their popularity and availability during project planning in 2018 and early 2019 and to provide a variety of sensor components and data processing methodologies for comparison.

Table 1, additional details in Appendix A. Many of the models evaluated are no longer available from the manufacturer because of significant changes or improvements to the technology made in the past five years. All devices used wall power except for three Clarity Node-S devices evaluated in Research Triangle Park, NC. Four of the sensors tested use different versions of the Plantower sensor (Nanchang City, China) while the other two use the Nova SDS011 (Nova Fitness Co., Ltd., Jinan, China) and the Particles Plus OPC (Stoughton, MA, USA).

### 2.3. Long-Term Monitoring Sites Selected

Seven colocation sites were chosen based on location, both geographic and relative to other selected sites, and the types of criteria pollutants being measured at each site (Table 2, Appendix A). These sites span five of the nine continental U.S. climate regions [20,21] (Figure 1), have a wide range of temperature, RH, and PM_2.5_ concentrations (Figure 2), and represent urban-to-neighborhood scale environments to ensure a variety of source influences. Sites outside of NC included Phoenix, Arizona (AZ); Denver, Colorado (CO); Wilmington, Delaware (DE); Decatur, Georgia (GA); Oklahoma City, Oklahoma (OK); Milwaukee, Wisconsin (WI). These are regulatory sites monitoring a variety of pollutants, except the NC site, which is the Air Innovation Research Site (AIRS) of the EPA that reports data to AirNow. Teledyne API T640 or T640x optical monitors (San Diego, CA, USA) were used for comparison at all sites except for West Phoenix, where a Thermo tapered element oscillating microbalance (TEOM) 1405-DF (ThermoFisher Scientific, Waltham, MA, USA) was used. All T640 and T640x data use the original firmware and are not reflective of the April 2023 firmware update. All monitors were maintained, and data was quality assured by local agency staff. Sensors were within 20 m horizontally of the monitors (as specified in the performance targets [10]), with many sensors within a few meters of the monitors.

### 2.4. Data Processing and Analysis

Most reference data were downloaded from the Air Quality System (AQS) (Appendix A). These data are quality assured and validated by the air agencies providing these data as specified in their site quality assurance project plan (QAPP) (e.g., flow and leak check instruments, investigating outliers, maintenance). NC data were provided by EPA staff directly after some quality assurance similar to other sites. Data not available from AQS were downloaded from AirNow Tech, including temperature and RH data from the AZ site. AirNow Tech data are not as closely quality-assured and quality-controlled as data from AQS.

Data were averaged to hourly averages. Plots were generated to visualize each month for each sensor type (Example: Appendix A) at each site. Each plot was visually inspected to identify common data issues.

For the CNO sensors, we evaluate both the raw uncalibrated PM_2.5_ values generated by the device and the values as corrected with the 2021 wildfire correction (CNO_wf) (Appendix A) (https://www.clarity.io/2021-wildfire-calibrations, accessed: 29 August 2024), now superseded by the PM_2.5_ Global Calibration v2 (https://www.clarity.io/blog/clarity-releases-v2-pm-global-calibration-model-with-significant-performance-improvements, last accessed 12 November 2024). For PAR, we used previously developed methods to exclude measurements when the A and B channel measurements are significantly different [22]. We also evaluated the performance using previously developed corrections (PAR_wf) [15,22]. The highest PAR hourly concentration was 448 µg/m^3^, so the extended correction for high concentrations (>570 µg/m^3^) was not needed [15]. PAR sensors were the only sensors with duplicate internal sensors (i.e., Plantower PMS5003s (Nanchang City, China)), labeled as channels A and B, evaluated during this study. Since other sensors did not have duplicate internal sensors, similar quality assurance could not be applied.

For this project, sites were visited roughly weekly, with some interruptions because of staffing and COVID-19. During these visits, the physical operation of the sensor was checked, data were physically downloaded from some sensors, and data were checked for completeness. A more thorough data review was completed later where research staff applied data flags to each data record. Many flags indicate why these data were not available, but some indicate abnormalities (e.g., sampling interval abnormality). Flag files were compiled for each sensor, documenting a variety of errors experienced during testing (Appendix A). These flags include sensors operating somewhere other than the field site, warm-up period, shutdown for data collection/maintenance, sensor maintenance, operator working near device, sampling interval abnormality, data loss due to user error, cellular/Wi-Fi communication error, power connection error, sensor malfunction either hardware or firmware, data incomplete due to meteorological, gas, or PM sensor malfunction, and data value issues including drastic/sudden spike or decrease, outside of expected range, or timestamp adjustments. Many of these flagged data were not removed (e.g., sampling interval abnormality). These were summarized to better understand common issues and failure patterns for the different sensor types. Since this paper focuses on PM_2.5_, only the PM_2.5_ sensor and “all” sensor flags were considered as, in some cases, the gas sensors failed separately (e.g., ozone (O_3_) flags not considered in this paper).

We also considered the influence of relative humidity by binning these data into 10% RH bins (e.g., 0–10%, 10–20%, and 20–30%) and considered the ratio of sensor PM_2.5_ divided by monitor PM_2.5_ in each bin to evaluate the percent RH Influence.(1)$$\mathrm{Percent}\;\mathrm{RH}\;\mathrm{Influence}=\frac{\mathrm{High}\;\mathrm{RH}\;\mathrm{Ratio}-\mathrm{Low}\;\mathrm{RH}\;\mathrm{Ratio}}{\mathrm{Mean}\;\mathrm{Ratio}}\;\times \;100\%$$
where High RH Ratio is the average ratio (hourly sensor/monitor) at the highest RH bin with at least 10 h of valid ratios between the sensor and the monitor, and Low RH Ratio is the ratio at the lowest RH bin with at least 10 h of valid ratios between the sensor and the monitor, and Mean Ratio is the mean of the means of all 10 bins. Note that the mean ratio is not the same as the average ratio of all data since there may be varying amounts of data in each bin depending on the environmental conditions during each collocation. Ratios were excluded if the monitor read < 5 µg/m^3^ as lower readings may be below the detection limit of the instruments, and small variations in the denominator may add noise obscuring the influence of RH.

## 3. Results and Discussion

### 3.1. Common Failure Points

Common points of failure seen across the units included loss of power due to loose or damaged power connections, battery issues, corrupted or damaged data storage (i.e., SD card, USB), and lost communication. Some delicate SD card ports were damaged due to the frequent data download schedule. Many of these failures could be addressed by modifications to the sensor design, including the use of connectors that prevent twisting of wires, avoiding silicone as the only means to secure fittings, mounting batteries in a way that prevents gravity from working against contact points, incorporating onboard data storage backup even if a sensor transmits data, testing sensor operation and data transmission in many different environments/countries, and adding visual status indicators for power, battery voltage, data logging, and data communications both on the sensor and in the online data dashboard. Real-time cellular communication strength indicators can help users site sensors in the field.

The most common flags for the CNO, ARS, AQY, and RAM sensors were sampling interval abnormalities (Appendix A). Each sensor model had expected sampling intervals (Appendix A), which varied by sensor type, except for the Clarity Node-S, which has a sampling period that is variable to accommodate the solar-powered operation. This flag was applied if intervals were skipped or if sampling did not conform to the expected interval. The biggest issue for PAR sensors was data loss due to a power connection error. For MAX sensors, the biggest issue was data loss from sensor hardware malfunction, most often associated with the battery. Other common sensor issues included PM sensor malfunction and data loss due to firmware malfunction, user error, or cellular/Wi-Fi communication error. Additional details are provided in the Appendix A.

### 3.2. Overall Performance by Site

First, we considered the performance of each type of sensor at each location without removing periods with sensor data issues. We evaluated performance based on the EPA Performance Target of R^2^ > 0.7 for PM_2.5_ [10]. We have not considered the additional performance metrics (e.g., slope and intercept) since it is assumed that an adequate R^2^ slope and intercept could also meet the performance targets with a basic linear regression correction. The performance target reports recommend evaluating at least a 30-day period [10], and most of the evaluations here cover more than a year. The performance targets also recommend using 24-h averages, though 1-h averages can be used as well [10]. We have used 1-h averages here due to the interest in high-time resolution data; however, 24-h averaged R^2^ would often be higher.

Sensor performance is highly variable by sensor type (Figure 3). The PAR sensors met the R^2^ target in all states except for OK, both with and without the wildfire correction (OK R^2^ = 0.67–0.68). CNO met the R^2^ target in all states except for NC and OK (R^2^ = 0.56–0.68) and, when the wildfire correction was used, no longer met the R^2^ target in GA (R^2^ decreased by 0.20) but performed better in OK (although the R^2^ only increased by 0.03). Similar to CNO, the MAX met the R^2^ target in all states except for NC and OK (R^2^ = 0.50–0.58). The AQY sensors met the EPA R^2^ performance target in WI and CO with weak correlations (R^2^ = 0.00–0.42) seen in other states, typically because of high concentration outliers. The RAM and ARS sensors performed poorly across all states (R^2^ = 0.00–0.57, R^2^ = 0.02–0.45, respectively). The RAM and PAR use the same internal sensor, the Plantower PMS5003; the difference in performance between these two sensors highlights the importance of sensor integration into the larger package (e.g., orientation, flow path, and onboard correction) and the impact on performance.

Sensor performance is highly variable by location (Figure 3) but some sites had adequate performance for most sensors. CO and WI had the most sensors meet the R^2^ target with six of eight sensors or corrected sensors meeting the target. This suggests that it may be easiest to use PM_2.5_ air sensors in CO and WI and obtain accurate results, likely because of the wider range of PM_2.5_ concentrations experienced, more consistent particle properties, and favorable meteorological conditions. DE and AZ had five of eight sensors meet the R^2^ target suggesting that PM_2.5_ sensors will typically perform well in these locations. AZ experienced dust impacts and has more large particles than other parts of the country (Appendix A). Many previous studies have shown PM air sensors often measure particles only 0.3 to 1 µm with variable success. This dust can lead to sensor underestimations and inaccuracies [23,24,25,26,27], which likely leads to poor or changing agreement between sensors and monitors in AZ. However, the R^2^ metric somewhat depends on the concentration range experienced, and AZ had many hourly PM_2.5_ concentrations above 100 µg/m^3^ (Figure 2, sample size (*N*) = 72) leading to typically higher R^2^ values (R^2^ = 0.00–0.92). In addition, these high-concentration events may be primarily wood smoke events instead of dust events (i.e., larger particle events) [28] that are easier for sensors to measure [1,23]. Sensors may have performed better in these locations for a variety of reasons.

Sensor performance has some dependence on the monitor used as a reference. Both the TEOM and T640/T640x were designated and operated as Federal Equivalent Methods. Past work comparing sensor performance to a TEOM and T640 has shown stronger correlations between sensors and the T640, likely because they are both optical methods, and have also shown significant fluctuations from the TEOM at 5 µg/m^3^ or less [29]. However, the T640 or T640x may provide slightly higher estimates of PM_2.5_ compared with the TEOM [30] but typically perform adequately [31].

Other sites had poor performance for half or more of the sensor types or corrections, indicating they may be more challenging environments for sensors to operate. GA had four of eight sensor or sensor corrections meet the target suggesting that most sensors may be able to provide adequate measurement accuracy in GA. Only two sensor types or sensor corrections (PAR and PAR wildfire corrected) met the R^2^ target in NC. All sensors were colocated in NC before and after testing, and typically, three sensors were run simultaneously throughout the project. Sensors have different responses (i.e., low precision), leading to the typically lower performance as measured by overall R^2^ in NC. This lack of precision has been shown in past work with similar sensors [12,32]. Only one sensor type or sensor correction (CNO_wf) met the R^2^ target in OK. The OK sites experienced changing particle size distributions (Appendix A) and properties. Changing particle size distributions can lead to sensor inaccuracies and variable relationships between the sensors and monitors [24,25,33], which likely leads to poor or changing agreement between sensors in OK.

Sites with typically better performance likely have more stable particle properties—other than AZ, which was previously discussed. In many cases, we would need additional information on particle size distribution and chemical composition to draw further conclusions about the differences at these sites. Many previous studies have been published looking into the PM_2.5_ characteristics at these sites and in these cities, showing variable particle properties (e.g., size distribution and chemical composition) depending on source [34], wind speed and direction [34,35,36,37,38,39], local meteorology [40], time of day [36,39,40,41,42], weekday versus weekend [43], and season [43]. These variations in particle properties can impact particle light scattering [44] and particle hygroscopicity [45], which can impact sensor performance [33]. However, it is unknown how relatable that work is to the period studied during this project. The time period during this study may differ because of year-to-year differences in sources and meteorology, spatially variable long-term trends in PM_2.5_ concentrations and compositions [46,47,48], and also because of the impacts of the covid pandemic on local and regional PM_2.5_ concentrations [49].

Our results are typically in line with past work for most types of sensors evaluated during this study. APT sensors (including the APT Maxima (MAX) and Minima) strongly correlate with reference measurements internationally [50,51,52,53], sometimes requiring an RH correction [54], sometimes requiring correction dependent on source composition [55], and exhibit a strong correlation in the lab [56,57] but correlations may be weaker in some locations and can be dependent on reference monitor type and data averaging [58]. To our knowledge, no work to date has directly compared the performance of the APT MAX and Minima, but the devices have the same internal components. CNO sensors have typically seen strong or near strong correlations (R^2^ = 0.69) with reference measurements internationally [59], although some have seen moderate correlations (R^2^ = 0.61) with limited improvement with correction [60]. Stronger correlations were observed when also correcting for temperature and RH [61]. PAR sensors are the most widely studied and have typically seen strong correlations across the U.S. and North America, especially once RH influences are accounted for [14,15,19,62], but have changing relationships during dust impacts [23,63]. Past PM_2.5_ evaluations of the ARS are not comparable due to changes in the internal sensing component [64]. To our knowledge, no past work has evaluated this model of the ARS, which incorporated the Particles Plus OPC. AQY has been strongly correlated with a Beta Attenuation Mass monitor in California during short-term evaluation [12], with mixed results during short-term smoke impacts [62], mixed results in California [65], and poor correlation in Texas [66]. More complicated network corrections have been proposed to improve AQY sensor performance for PM_2.5._ However, these corrections are dependent on external information (i.e., using monitor data as a proxy) and not just data coming from the sensor itself [65]. Many of the international evaluations have higher average PM_2.5_ concentrations and wider ranges or PM_2.5_ potentially contributing to stronger correlations in those areas. Additional correction could further improve the performance of these sensors.

Past work has typically shown better performance for the RAM than was found during this study. The RAM has shown strong correlations during short-term smoke impacts [62] and mixed but near-adequate results internationally [67]. It is unknown how similar the correction that came with our sensor was to those used in these past projects. Much past work with the RAM used a previous design where they were attached to external PM_2.5_ sensors (e.g., Met-One Neighborhood Particulate Monitor) or deployed alongside PAR sensors [68]. It is likely that changes to the sensor design over time have contributed to the differences in results between this study and past work.

### 3.3. Common Data Issues

Four common issues were identified in the monthly plots and are outlined in the sections below, along with an additional section on RH influence. The four common issues identified were repeat zero measurements, single point high concentration outliers, baseline shift where the relationship between the sensors and monitor changes for a period of hours, and variable relationships between sensors and monitors where the relationship between sensors and monitors changes for longer periods (e.g., days). These issues were removed before considering the influence of RH and variability in the bias sections below but were not removed before considering the overall performance (Section 3.2 above).

#### 3.3.1. Zero

Some of the sensors reported repeat zeros. This was especially common for the ARS sensors, and often this was an indicator that the PM sensor pump had failed. All zeros were removed from these ARS data (12%), and in other sensors, short periods with repeat zeros were removed (<1%). These short periods of zeros were identified visually. It is important to remove zeros carefully from the dataset as past work has also shown some sensors will read repeat zeros when the concentrations detected are low and near the limit of detection of the sensor [69]. Depending on the project objectives, zeros when the concentrations are low and near the limit of detection would be kept in the dataset so as not to bias the dataset high. For example, Figure 4 shows three MAX sensors operating in NC. In late October, one of the MAX (purple) starts reporting repeat zeros; however, zeros also occur when concentrations are low, as shown by other sensors in the time series and scatter plots.

#### 3.3.2. Outlier

Some sensors also experienced outliers where these sensor data would suddenly be 10s of µg/m^3^ higher than the monitor (e.g., Figure 5A,B) or previous and subsequent 1-h sensor readings. However, other examples of outliers are true short-term PM pollution events that would be hard to identify without reference data (e.g., Figure 5C,D). Both figures show a concentration jump of about 50 µg/m^3^ compared with the average 1-h average concentrations over the prior week. Without additional details about what might be causing a 1-h high concentration event and data from the monitor, it would be hard to identify whether the sensor readings represent outliers representative of a sensor issue or a real pollution event. PAR is unique as its dual Plantower design allows for most sensor malfunction outliers to be excluded since it is highly unlikely both channels will have outlier issues at the same time. If both sensors report a high concentration, it is more likely a real PM_2.5_ event.

While more time could be spent developing mathematical criteria to exclude outliers from our dataset, it is important to consider that colocation often occurs at regulatory sites away from localized sources, while sensor networks are often deployed to measure pollutant hotspots [70,71]. This means any methods validated at a regulatory site might incorrectly remove outliers from sensors deployed in a network with different PM patterns (e.g., localized sources and localized geography). For this reason, outliers were removed from this colocation study dataset manually by visually inspecting each monthly plot.

#### 3.3.3. Baseline Shift

Sometimes sensors saw baseline shifts leading to short-term strong overestimates of PM_2.5_ compared with the monitor (Figure 6A,B). Baseline shift errors are times when the sensor strongly overestimates PM_2.5_ concentrations more than usual for a particular sensor and for more than a single point outlier, as described in the section above. However, PM_2.5_ sometimes has true baseline shifts in concentration when regional or long-range pollution blows in (Figure 6C,D). An example of a real baseline shift was experienced during late June of 2020 when Saharan dust impacted the United States [72]. Similar to the outlier example, without additional information on local or regional sources, air monitoring data, or data from colocated sensors, it would be hard to identify baseline shifts from those due to true PM_2.5_ events. We define these events differently from the outliers, as multiple points in a row are impacted instead of a single point.

#### 3.3.4. Variable Relationship Between Sensor and Monitor

While some months showed strong correlations between sensors and monitors, some showed weak correlations or distinctly different relationship patterns during other months. Both examples in Figure 7, panels B and D, show two distinct prongs in the scatter plots. Panels A and B in Figure 7 show data from two colocated RAM sensors along with reference data. In this case, the response of one sensor changes mid-month, showing higher concentrations that agree more closely with the reference monitor. This changed response cannot be explained. The example shown in panels C and D in Figure 7 is from the Saharan dust event that occurred late in the month. In this case, CNO often reads higher than the monitor (i.e., as shown in the first part of the time series through 22 June and with many of the scatter plot points above the 1:1 line). However, on 26–28 June, the sensor did not detect the dust particles, while the reference monitor detected dust particles, leading to a relatively low sensor response.

### 3.4. RH Influence

Much past work has documented that sensors are often biased by high RH [16,33,68]. Some sensors, including the ARS, CNO, MAX, and PAR, show increasing overestimation of PM_2.5_ as RH increases (Figure 8). AQY and RAM show little change in bias across different RHs. After applying the wildfire corrections, both PAR and CNO show little impact of RH on bias. The RH correction in the CNO and PAR wildfire equations are similar (CNO = −0.0510 × RH, PAR = −0.0862 × RH) and appear effective across all sites. However, these terms are not necessarily directly comparable because of the differences in other terms in the equation, potential differences in RH measurements, and other factors.

To compare the magnitude of the influence of RH by site and sensor, we compared the change in the mean sensor to monitor concentration ratio at low and high RH. Figure 9 expresses the influence of RH as a percent, calculated by dividing the difference in the mean ratio in the highest and lowest RH bins by the mean ratio across all bins. AQY is well corrected for RH across sites, with the influence at most sites within ±35%. The ARS has the largest influence from RH, with measurements at high RH 200% higher or more at most sites; this is the only device that uses an OPC which may lead to the larger influence of RH [33]. CNO and MAX see overestimations due to RH at all sites, while PAR sees overestimation at most sites. Negative RH influence, below 0 on the plot, indicates over-correction for RH since we would expect RH to increase the size of the particles and, therefore, the PM_2.5_ estimates. After correction, PAR_wf and the RAM over-correct for RH at most sites, while CNO_wf slightly over-corrects. The influence of RH on the measurements is dependent on the sensor, with large differences seen across sensor types.

The difference in RH influence across locations is more similar than the difference in influence by sensor type. Typically, higher RH leads to higher sensor PM_2.5_ estimates, but some sensors at each site show the opposite trend (i.e., higher RH leads to lower PM_2.5_ estimates), suggesting that measurements have been over-corrected. Some of the difference in RH influence between locations could be due to variations in particle type and hygroscopicity but some of the variation may be due to differences in individual sensor performance.

The federal equivalent method keeps the RH of the particles relatively constant by conditioning the sampled aerosol. None of the sensors evaluated in this study had driers or RH controls. If sensors do not physically control RH, correction algorithms are typically needed to account for water associated with the particles. These corrections allow the measurements from sensors to be comparable to federal equivalent methods that control RH. These algorithms are dependent on the RH measured by the sensor (e.g., CNO wildfire correction and PAR wildfire correction), so it is important to understand the accuracy of the RH measurement as well. Scatterplots in Figure 10 explore the agreement between sensor-based RH measurements compared with the high-quality measurements made at each air monitoring site. Some of the scatter in these plots may be due to the difference in the internal operating temperature of the sensor and fluctuations as the sensor experiences shade versus the sun. Typically, these sensors would experience periods of sun and shade every day. In some cases, the sensors may be measuring the internal RH as higher than what the particles experience as the ambient RH making these values potentially more useful for correction. The ARS RH sensors seem to be the least consistent and reliable. Some of the AQY RH measurements are at 100% even when ambient RH is low (~25%). Some time periods are identified for PAR and RAM where stuck values occur, indicating sensor or communication error.

### 3.5. Variability in Bias

#### 3.5.1. Bias by Sensor and Location

We considered mean bias error (MBE) by sensor make and location (Figure 11). These MBEs are comparable because the average concentrations, as measured by the monitors, were 7–10 µg/m^3^ for each location. AQY and the RAM strongly underestimate PM_2.5_ across all locations. The ARS has wide variability, with some sensors showing strong underestimations and some sensors showing strong overestimations. Without correction, CNO and PAR overestimate the concentrations at multiple sites. CNO_wf, MAX, and PAR_wf typically have low biases within ±1.7 µg/m^3^ (20% of the average PM_2.5_ concentration). When comparing bias by site, the differences are less distinct, with all sites having at least one sensor with a bias of less than 1.7 µg/m^3^ and other sensors that strongly over or underestimate. The typical bias is more variable by sensor type, with less difference seen in bias by location.

#### 3.5.2. Hour of Day Performance

PM_2.5_ concentrations vary over the day and by site in terms of average concentration and variability (Figure 12 and Appendix A). AZ has the largest daily variability in average PM_2.5_ by hour, with concentrations varying by more than 6 µg/m^3^, and WI has the least variability with less than 1 µg/m^3^ difference over the day. Although already corrected, both CNO_wf and PAR_wf data typically underestimate PM_2.5_ concentrations (Figure 12). This may be due to the comparison with T640 and T640x data which have been shown to be slightly biased high [15,30,73,74,75]. While a new correction for T640 data was developed that slightly adjusts the PM_2.5_ measurements (https://downloads.regulations.gov/EPA-HQ-OAR-2023-0642-0029/content.pdf, accessed on 3 September 2024), these data were collected prior to the release of the correction and so uncorrected T640 data have been used throughout this paper. If the T640 measurements were adjusted for the 2 µg/m^3^ overestimation, the bias for the sensors would be closer to zero. Interestingly, the CNO wildfire correction works almost perfectly in WI with little daily variation in bias, which remains near zero. For PAR, the bias is similar (about 1 µg/m^3^) for all sites except for AZ and CO, and all sites are slightly more biased in the morning and during higher RH. In some cases, the sensors do not show the same daily patterns. In GA, NC, and OK, the monitor shows lower evening concentrations, but GA CNO_wf, both NC sensor types, and the OK CNO_wf show higher evening concentrations. These sensors could be unreliable in providing information on what time of day the air is cleanest to exercise or spend time outdoors. This may be due to differences in particle properties or environmental conditions at different times of the day, leading to different sensor responses. These results suggest more work may be needed to improve the corrections to have stable bias by the hour for each day.

#### 3.5.3. Monthly Bias

Bias is variable by month (Figure 13, Table 3) with bias from +13 µg/m^3^ (DE PAR July 2019; NC PAR Oct 2020, WI CNO Dec 2010) to −18 µg/m^3^ (AZ RAM Jan 2020) (additional details Appendix A). The most variation in MBE occurs in NC for the ARS, CNO_wf, MAX, and PAR; the variability is due to both seasonal differences in performance and the difference in bias between duplicate sensors operated in NC (i.e., spread in MBE values in the same month) indicating low precision. Excluding NC, the ARS, CNO, and MAX have the most monthly difference in WI, and CNO_wf, PAR_wf, and RAM have the most monthly difference in MBE in AZ. AQY has the most variability in MBE in OK, and PAR has the most variability in DE. On the flip side, most sensors had the least variability in MBE in GA (CNO, MAX, PAR, PAR_wf, and RAM), with AQY and ARS having the least variability in CO and CNO_wf having the least variability in WI. These results suggest that WI and AZ may have more seasonal differences in particle properties and environmental conditions that lead to variable performance. However, corrections such as the CNO_wf correction may be able to account for much of the difference in monthly bias in WI due to RH influences. In AZ more dust events occur in the summer [76] and more smoke in the winter [28], likely leading to variable performance. AZ has the largest change in PM_2.5_ concentration over months of the year with a minimum of 5–6 µg/m^3^ in spring and summer months (i.e., August and Sept 2019 and March–July 2020) and a maximum of 30 µg/m^3^ in Dec of 2020 and 22 µg/m^3^ in Jan 2020. In AZ, some sensors see larger underestimation (i.e., negative MBE) at higher concentrations (e.g., AQY, CNO_wf, PAR_wf, and RAM) while others see a larger overestimation at higher concentrations (e.g., CNO, MAX, and PAR) which seems mostly dependent on the different concentration ranges experienced. For example, much higher concentrations experienced in AZ during Jan of 2020 than July 2020 leading to larger over and underestimations depending on concentration.

Even after applying corrections, PAR_wf and CNO_wf show strong biases at different sites across different months. While the median MBE is −1 µg/m^3^ for CNO_wf, the MBE varies from −7 to +1 µg/m^3^. For PAR_wf, the median MBE is also −1 µg/m^3^, but the MBE varies from −9 to +2 µg/m^3^. It is important to understand the limitations of sensors and the potential for seasonal bias. Without careful correction, conclusions about seasonal patterns in PM_2.5_ as measured by sensors should be used with care, as differences in particle properties and environmental conditions may lead to incorrect conclusions. In addition, different sites may need different lengths of colocations to generate useful results, and leaving a sensor colocated will help determine how sensor performance may change seasonally.

We also compare the range in monthly MBE to the overall R^2^ for each sensor type at each site (Table 3). The R^2^ values are calculated after removing the common data issues identified in Section 3.3, so in many cases, they are higher than those shown in Figure 3. The median range of MBE for each site/sensor model is 7 µg/m^3^. In AZ and OK, sensors typically have large ranges in monthly MBE. In AZ, the R^2^ target is typically met, while in OK, it typically is not. It may be challenging for most sensors to accurately measure PM_2.5_ in AZ and OK because of changing particle properties and seasonal differences. In GA and NC, many sensors do not meet the R^2^ targets but have a low range of MBE values. It may be more challenging for sensors to measure accurate concentrations in GA and NC as well. In CO, DE, and WI, the range in MBE is low, and the R^2^ target is met by most sensors indicating it may be easiest for sensors to perform in these environments.

## 4. Conclusions

Long-term air sensor evaluations across six states highlighted common failure points for air sensors, including both physical (e.g., shipping damage, communication loss, and wiring coming unplugged) and data issues (e.g., sampling frequency issues, outliers, stuck zero, baseline shifts, and variable relationships between sensor and monitor PM_2.5_). Many of these data issues would be hard to identify without colocated monitor data or at least data from a duplicate sensor.

RH can lead to strong bias from air sensors, highlighting the need for physical humidity control (e.g., dryer) or robust correction algorithms and performance evaluation of the onboard RH measurement. Sensors with stronger RH influences (e.g., OPCs) may benefit the most from nonlinear RH corrections. Common RH correction methods may reduce bias on average, as shown with AQY, CNO_wf, and PAR_wf. However, care should be taken to ensure these data are not over-corrected.

Bias may be variable by month, indicating that seasonal corrections may improve performance and highlighting the need to consider sensor limitations when drawing conclusions (e.g., ensure seasonal sensor bias is limited before drawing conclusions about seasonal PM_2.5_ concentrations using sensors).

The sensors evaluated show varying degrees of promise to provide accurate PM_2.5_ data across the U.S. AQY could be accurate if baseline shift issues can be improved or identified and removed. The ARS will be challenging to use because of the low R^2^, high RH influence, frequent repeat zeros, and the largely variable seasonal bias. The RAM will be challenging to use because of the low R^2^, which may be due in part to the over-correction for RH influence. CNO, CNO_wf, PAR, PAR_wf, and MAX could perform accurately in many locations. The performance of all sensors could be improved with further individual sensor-specific or location-specific corrections in some cases, including improved RH correction.

Much improvement has occurred in the field of air sensors since the purchase of these sensors five years ago. It is largely unknown how the performance of these devices will compare to the latest versions from each manufacturer because of changes in quality assurance and data processing practices. Many sensors on the market today still use similar internal sensors (e.g., Plantower), but some of the issues identified in this paper may have already been corrected by better manufacturing, more sophisticated software, advancements in calibration, and the introduction of improved automated quality control.

Sampling frequency may also impact the comparability of these results. For example, most of the CNO sensors evaluated were the CNO Node with wall power that typically sampled more than 20 times per hour. However, this model has been discontinued, and now all CNO Node-S sensors use solar power. This device provides hourly averages based on fewer measurements per hour. This may lead to greater uncertainty, lower internal temperatures, different RH influences, and overall different performance.

While this study captures some of the wide variety in sensor performance results to date, the results of this study may not apply to performance across all parts of the U.S. with different conditions. In addition, other parts of the world may have different local meteorological conditions, particle properties, and PM concentrations, leading to different sensor performances. Though we captured a year or more of data at these sites, it does not represent the full range of conditions that could be experienced as some events (e.g., extreme wildfire smoke) may only happen every few years.

Additional analysis could be accomplished with this dataset. By making it publicly available, we hope others will use it to support ongoing work to understand air sensor performance better, draw more conclusions, and validate correction methods under development.

## Figures and Tables

**Figure 1 sensors-25-01265-f001:**
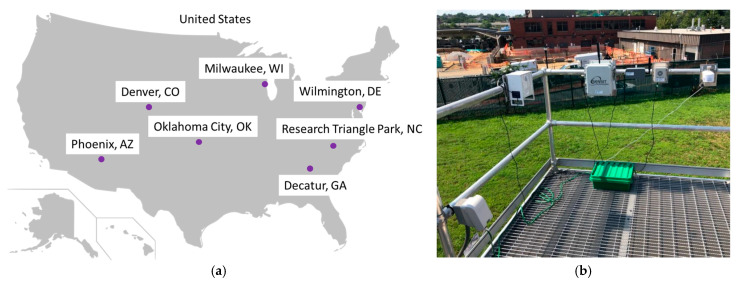
(**a**) Map of Selected Regulatory Monitoring Sites and (**b**) DE Site—Deployed Sensors. On railing: AQY, RAM, CNO, MAX, PAR (ARS deployed later and not pictured).

**Figure 2 sensors-25-01265-f002:**
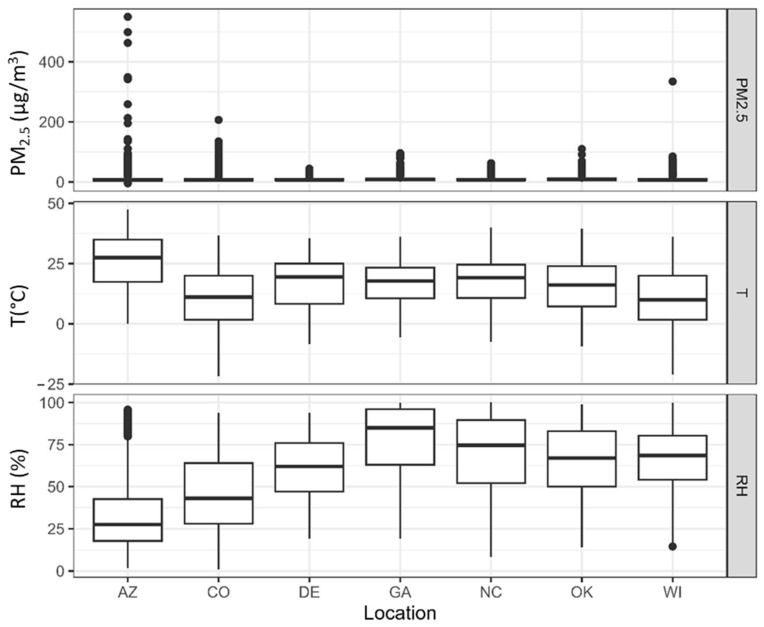
Boxplot showing the ranges of PM_2.5_ (µg/m^3^), T (°C), and RH (%) experienced at each site based on the reference measurements. These values cover the colocation period, typically July or August 2019 until Oct 2020 or January 2021, depending on the site and sensor types (additional details in the Appendix A).

**Figure 3 sensors-25-01265-f003:**
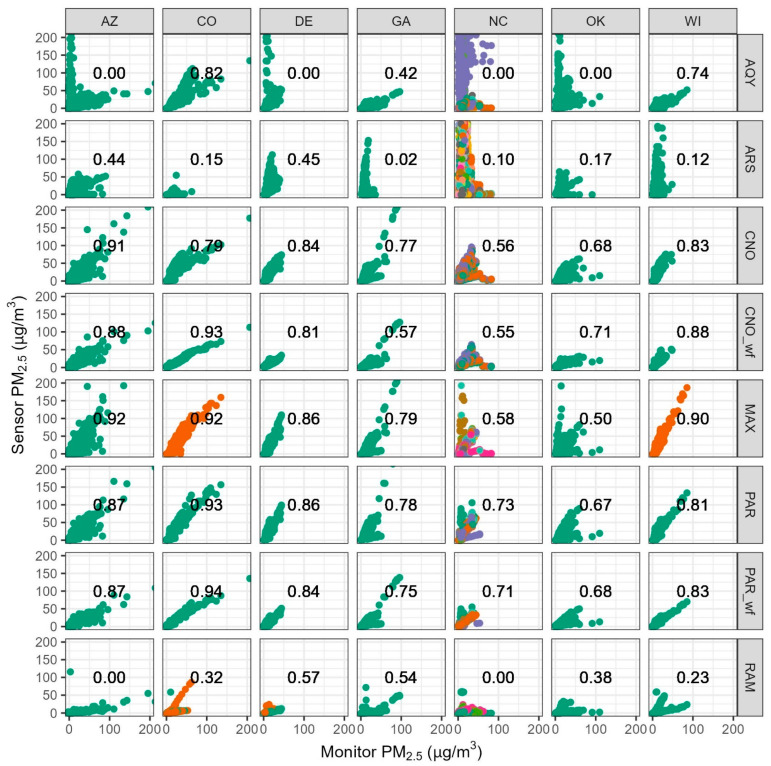
Hourly averaged sensor versus reference for all sensors (colored by unit ID) before sensor problems are removed with R^2^ in the center of each plot. Points above 200 µg/m^3^ are excluded from the plot to improve visualization (but were left in for all analyses).

**Figure 4 sensors-25-01265-f004:**
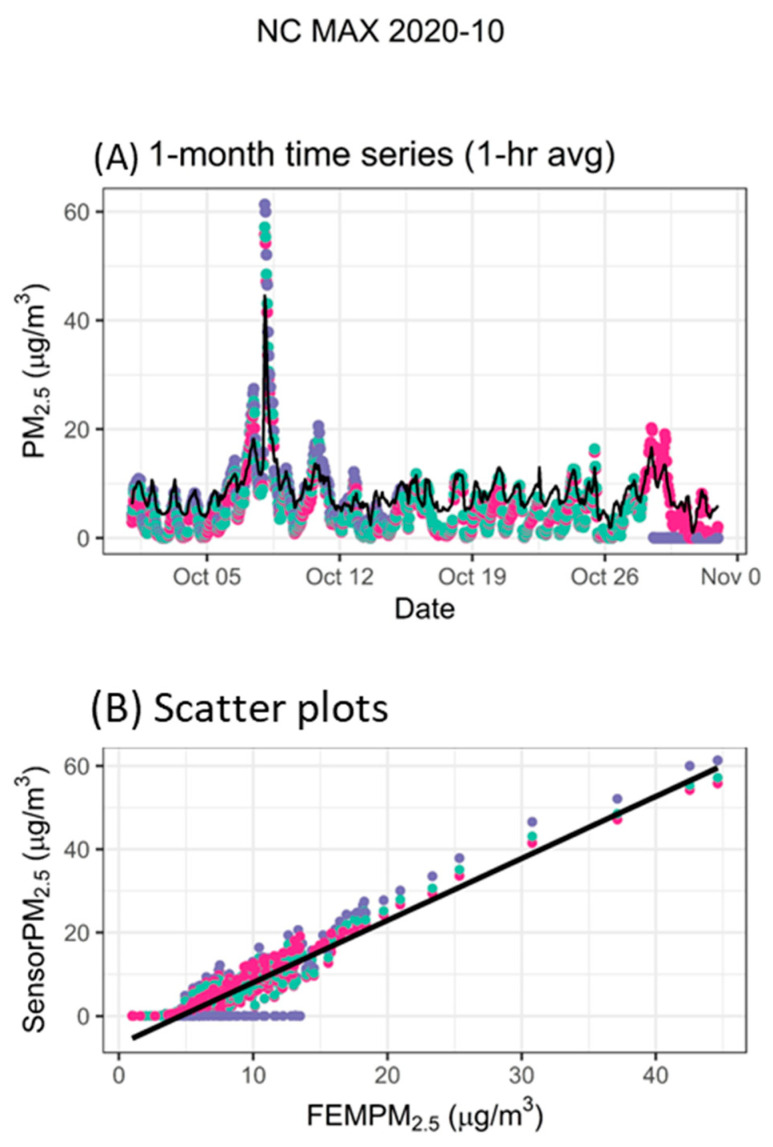
Example of repeat zeros is shown by the purple MAX sensor (different colors represent different sensors) compared with the monitor (black line on time series).

**Figure 5 sensors-25-01265-f005:**
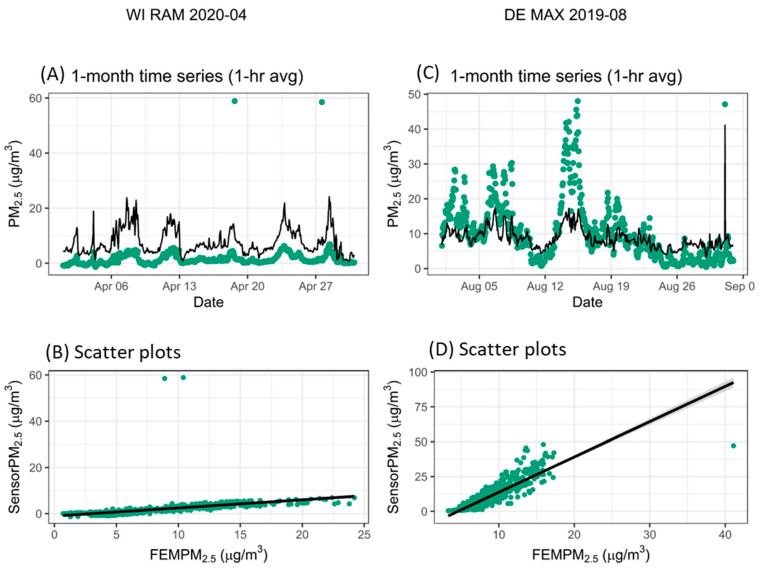
Example data showing sensor outliers where (**A**,**B**) show data outliers from RAM sensors in WI while (**C**,**D**) show a real concentration event measured by the MAX in DE. The black line on the time series is the monitor and the green is the sensor.

**Figure 6 sensors-25-01265-f006:**
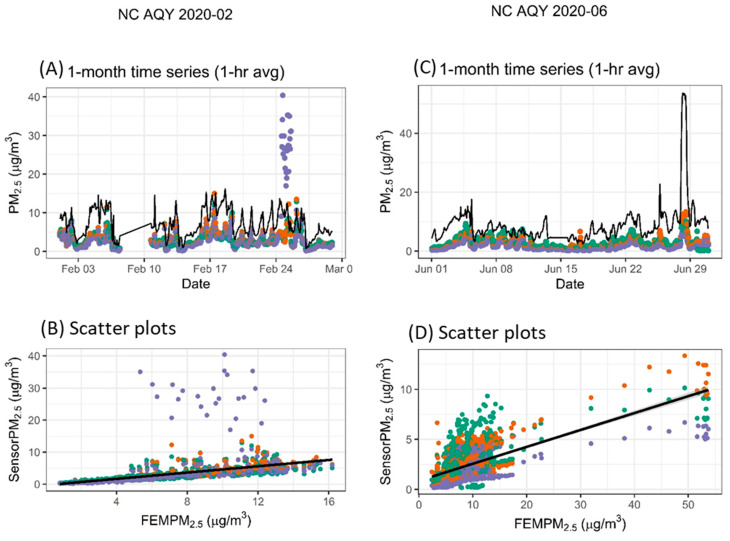
Examples of a baseline shift where one sensor suddenly sees concentrations 20–40 µg/m^3^ higher than the monitor (black line on the time series) (**A**,**B**) and an example where the monitor sees a real baseline shift in PM_2.5_ concentrations of more than 40 µg/m^3^ (**C**,**D**) because of a Saharan dust event. Colors indicate different sensors.

**Figure 7 sensors-25-01265-f007:**
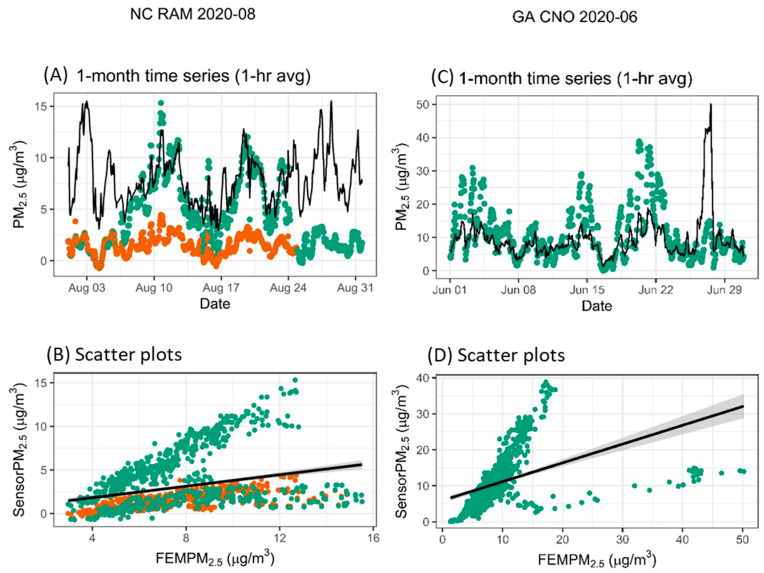
Two examples showing variable but distinct relationships between the sensors and the monitor (black line on the time series). Two sensors are included in the NC RAM example indicated by different colors.

**Figure 8 sensors-25-01265-f008:**
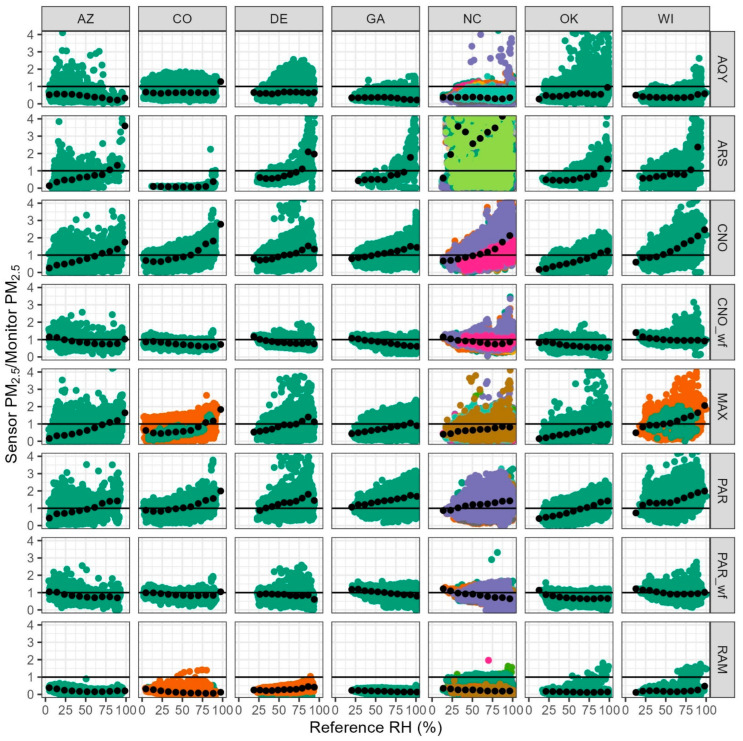
Scatter plot showing the influence of RH on the ratio of PM_2.5_ sensor/PM_2.5_ monitor. Hours where the monitor PM_2.5_ is less than 5 µg/m^3^ have been excluded. Colors indicate unit ID and black dots show the average ratio in each of the 10 bins (e.g., 0–10% and 10–20%).

**Figure 9 sensors-25-01265-f009:**
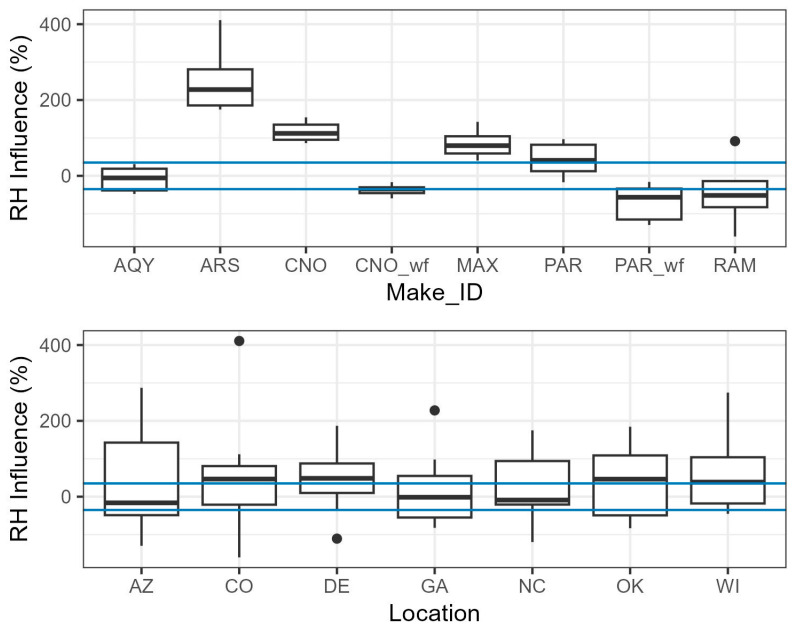
Boxplots showing the influence of RH with more variation by sensor make than by location. Each point represents the relative difference in the ratio of sensor/monitor from high to low RH, as shown in Figure 8. The area between the blue lines at ±35% indicates weak RH influence.

**Figure 10 sensors-25-01265-f010:**
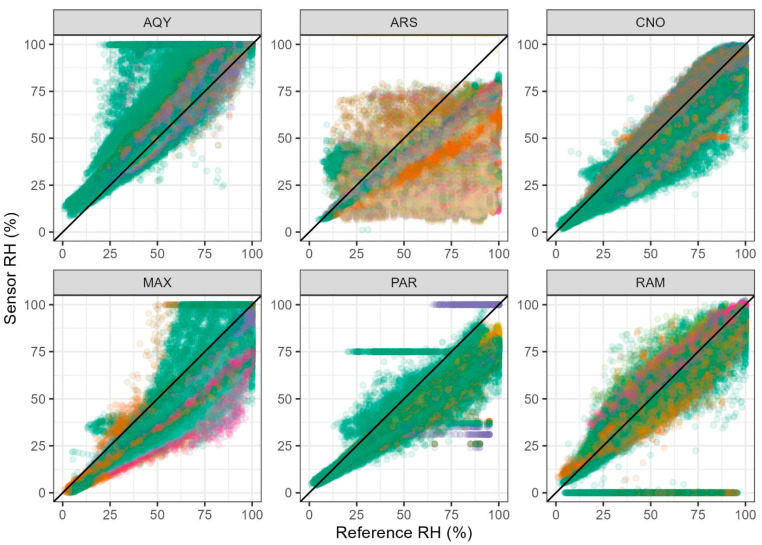
Differences in the hourly RH measured by the sensor compared with the independent reference RH at each site. Colors indicate different sensors, and the black line is the 1:1.

**Figure 11 sensors-25-01265-f011:**
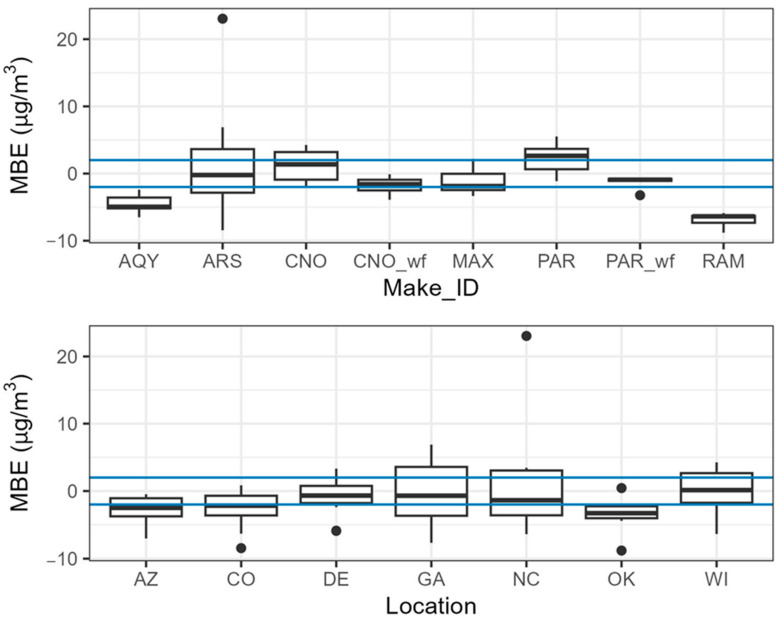
Bias by sensor make and location. The area between the blue lines (±1.7 µg/m^3^) indicates low bias (20% of the average concentration).

**Figure 12 sensors-25-01265-f012:**
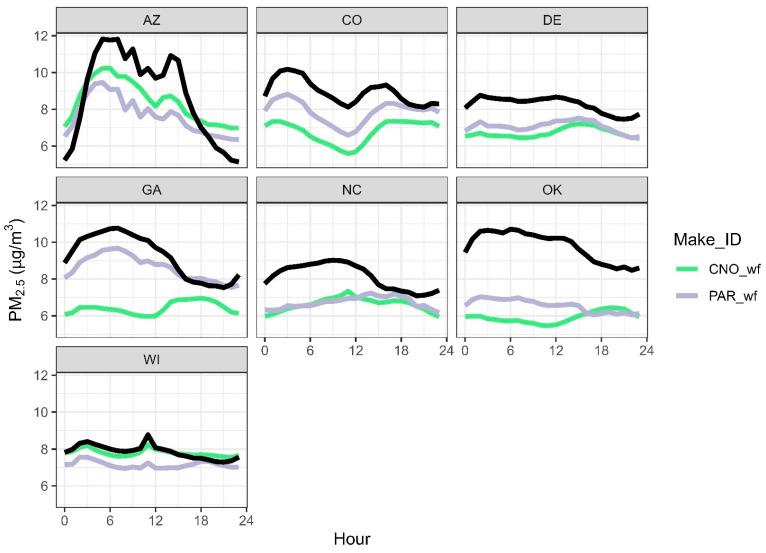
PM_2.5_ concentrations by location and hour of day. The monitor is in black.

**Figure 13 sensors-25-01265-f013:**
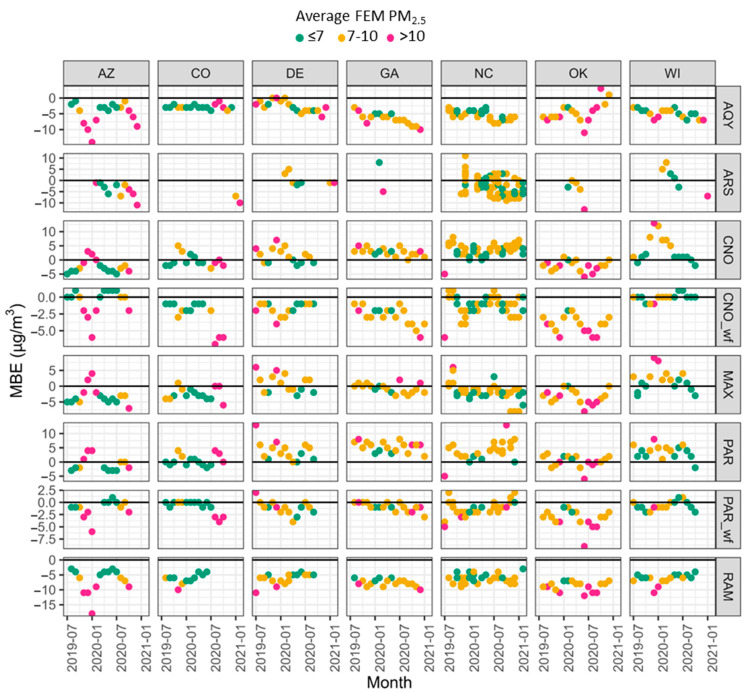
Mean Bias Error (MBE) by month across all sites and sensor types. Multiple points per month in NC due to multiple sensors running simultaneously. Note the variable *y*-axis. The black horizontal line on each plot indicates MBE = 0 µg/m^3^.

**Table 1 sensors-25-01265-t001:** List of sensors evaluated see additional details in Appendix A.

						Number Evaluated			
ID	Make	Model	Internal PM Sensor	Communication	Power Source	NC	Other Sites	Measured Pollutants	Sampling Interval
AQY	Aeroqual (Auckland, New Zealand)	AQY *	Nova SDS011	Cellular Wi-Fi (NC only)	Wall	3	6	PM_2.5_, NO_2_, O_3_, T, RH	1 min
CNO	Clarity Movement Co. (Berkeley, CA, USA)	Node *	Plantower PMS6003	Cellular	Wall	-	6	PM_2.5_, NO_2_ *, T, RH	~5 min (Node) ~15 min (Node-S, NC only)
		Node-S		Wi-Fi	Solar	3	-	PM_2.5_, NO_2_ *, T, RH	30 s
MAX	Applied Particle Technology (Boise, ID, USA)	Maxima	Plantower PMSA003	Wi-Fi	Wall	3	6	PM_1_, PM_2.5_, PM_10_, T, RH, P	30 s
PAR	PurpleAir (Draper, UT, USA)	PA-II-SD *	Plantower PMS5003 (×2)	Wi-Fi	Wall	3	6	PM_1_, PM_2.5_, PM_10_, T, RH, P	2 min
RAM	Sensit Technologies (Valparaiso, IN, USA)	RAMP	Plantower PMS5003	Direct (no Wi-Fi/Cellular)	Wall	3	6	PM_2.5_, CO, NO, NO_2_, SO_2_, O_3_	15 s
ARS	Aerodyne ‡ (Billerica, MA, USA)	Arisense *	Particles Plus OPC	Cellular	Wall	7	6	PM_1_, PM_2.5_, PM_10_, CO, CO_2_, NO, NO_2_, O_3_, T, RH, P, WS, WD	2 min

* These make/models are no longer available from the manufacturer. ^‡^ Devices purchased from Aerodyne (Billerica, MA, USA), but then the company spun off into QuantAQ (Somerville, MA, USA).

**Table 2 sensors-25-01265-t002:** Selected monitoring sites and comparison monitors. The reported monitor average and maximum concentrations are for the duration of colocation.

Location (City, State)	AQS ID	Monitor *	Spatial Scale	Site Type	Average Monitor PM_2.5_	Maximum Hourly Monitor PM_2.5_
					(µg/m^3^)	(µg/m^3^)
Phoenix, AZ, USA	04-013-0019	Thermo TEOM 1405-DF	Neighborhood	Population Exposure Highest Concentration	8.9	550
Denver, CO, USA	08-031-0026	Teledyne T640	Neighborhood Urban	National Core Network (Ncore) State or Local Air Monitoring Stations (SLAMS)	8.8	207
Wilmington, DE, USA	10-003-2004	Teledyne T640	Neighborhood	Population Exposure Maximum Concentration NCore Photochemical Assessment Monitoring Stations (PAMS)	8.3	44
Decatur, GA, USA	13-089-0002	Teledyne T640x	Neighborhood	Population Exposure Highest Concentration	9.1	96
Research Triangle Park, NC, USA	37-063-0099	Teledyne T640	Neighborhood	NCore	8.2	82
Oklahoma City, OK, USA	40-109-1037	Teledyne T640 (until 31 December 2019) Teledyne T640x (starting 1 January 2020)	Urban Population Exposure	SLAMS	10.0	110
Milwaukee, WI, USA	55-079-0026	Teledyne T640x	Urban Neighborhood Population Exposure	SLAMS	7.9	335

* All T640 and T640x data are not reflective of the April 2023 firmware update that implemented the alignment factor.

**Table 3 sensors-25-01265-t003:** Summary of range of monthly MBE (max MBE–min MBE) by sensor make and location. Shaded cells are locations where the most variation by sensor type is seen, and shaded R^2^ < 0.7 does not meet the performance target. Statistics are calculated after removing the common data issues identified in Section 3.3, so results differ from Figure 3.

				Uncorrected PAR, CNO	
Make ID	Location	Range MBE	R^2^	Range MBE	R^2^
RAM	AZ	15	0.88		
AQY	AZ	13	0.64		
MAX	AZ	11	0.92		
ARS	AZ	10	0.45		
PAR_wf	AZ	7	0.87	7	0.87
CNO_wf	AZ	7	0.88	8	0.91
MAX	CO	7	0.92		
RAM	CO	6	0.35		
CNO_wf	CO	6	0.93	8	0.79
PAR_wf	CO	4	0.94	6	0.93
ARS	CO	3	0.15		
AQY	CO	3	0.82		
MAX	DE	9	0.86		
ARS	DE	7	0.45		
RAM	DE	7	0.58		
AQY	DE	6	0.75		
PAR_wf	DE	6	0.84	13	0.86
CNO_wf	DE	3	0.81	9	0.84
ARS	GA	13	0.12		
AQY	GA	7	0.42		
CNO_wf	GA	5	0.57	5	0.77
MAX	GA	5	0.79		
RAM	GA	4	0.64		
PAR_wf	GA	3	0.75	6	0.78
ARS	NC	20	0.12		
MAX	NC	14	0.67		
CNO_wf	NC	7	0.6	13	0.61
PAR_wf	NC	7	0.76	18	0.77
RAM	NC	6	0.32		
AQY	NC	5	0.41		
AQY	OK	14	0.21		
ARS	OK	13	0.42		
MAX	OK	8	0.55		
PAR_wf	OK	8	0.68	9	0.67
RAM	OK	5	0.38		
CNO_wf	OK	4	0.71	7	0.68
ARS	WI	15	0.16		
MAX	WI	12	0.9		
RAM	WI	7	0.45		
AQY	WI	4	0.75		
PAR_wf	WI	3	0.83	10	0.81
CNO_wf	WI	2	0.88	15	0.83

## Data Availability

Data will be available after publication from DOI: 10.23719/1531918.

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
