# Peer review of "Evaluation of Long-Term Performance of Six PM_2.5_ Sensor Types"

_sensors, 2025, doi:10.3390/s25041265_

Round 1

Reviewer 1 Report

Comments and Suggestions for Authors

Reviewer comments, manuscript sensors-3412783 by Barkjohn et al. "Long term performance of six PM2.5 sensors across the United States ".

Low-cost sensors provide real-time in situ data on air quality in the regions where the monitoring infrastructure is limited. In spite of their lower quality, they are still valuable but require calibration and intercomparison with qualified data sources. There are a lot of papers published currently on this topic (see reference in the manuscript as well), but the presented work is useful due to the more or less long duration of comparison that helps to understand the stability of the sensors. The choice of sensors (mostly Plantower) is positive because these sensors are stable and reliable compared to others. That study is significant for understanding the low-cost monitors' performance, stability, measurement's reliability and providing advice for improving received air parameters data. The manuscript presents novel results that are useful for validating data quality assurance, correcting the choice of air sensors for air quality monitoring, and methods for improving sensor performance.

General comments

The investigation and comparison of the low-cost air quality PM2.5 sensors to examine their accuracy and precision to improve the reliability of data. The authors investigate six types of air sensors at several air quality monitoring sites in different aerosol contamination and relative humidity conditions. 

The topic of this manuscript is relevant to the field, interesting and essential, and the manuscript is in the scope of the Sensors journal.

Last time there are many different types of low-cost sensors available for air quality control (e.g., AirVisual Pro, Temtop, Plantower sensors, other). Therefore, comparison of sensors fills the gap of knowledge which sensor is relevant and qualitative for use.

The conclusions are consistent in general with the results presented and address the main objectives of the study. However, the conclusions should be more specific and focused on research tasks like identifying common failure reasons of sensors, clarifying the influence of relative humidity, and proposing methods to reduce this influence.  The manuscript could be also updated in the Conclusion section with recommendations on which type of sensors is most appropriate for qualitative measurements. 

The reference list covers a sufficient number of papers on the topic of the manuscript published recently. However, some informative papers are still missing. For example, Karagulian et al. https://doi.org/10.3390/atmos10090506 presented a comprehensive review of low-cost air quality sensors. 

The figures are of bad quality, so almost all figures should be re-drawn. Need to reduce the size of ''dots'' at least in two times. Reduce width of lines in plots, make figures readable to see lines/dots variations. The Figures in the Supplement Information document should be improved as well.

Specific comments

Title: should be corrected like e.g., ''Long term performance of six PM2.5 sensors across the United States'' ----> ''Evaluation of long-term performance of six low-cost PM2.5 sensor types''

I am not sure that geographic information is essential to be mentioned in the Title

Abstract.

Line 22: explain the abbreviation EPA

L23: ''This project'' - what ''project'' you mean?

Keyword: remove ' Evaluation'' after title correction

L83: SI) ---> Supplementary Information (SI)

L61: ''This project'' - what ''project'' you mean?

Figure 1: The dots on the plots are too large and should be re-plotted. Needs to reduce the size of ''dots'' at least in two times. The same for Figures: 2, 3, 4, 5, 6!!!, 7!!!, 9,

Reduce width of lines in plot in Figure 8,

The same corrections need for Figures in SI. Make figures readable to see variations in lines/dots.

L417-418: sentence is not clear, should be re-written.

Technical comments:

In all text, a hyphen should be used instead of a dash, and a minus sign should be used instead of a dash. Need to make a difference between dash -, hyphen –, and minus sign − where appropriate. Check and correct through the text.

Final conclusion

The manuscript includes essential and new data and results to understand better the reliability of air quality (PM2.5) sensors for atmosphere pollution control. The manuscript can be published after minor revision, update the conclusion section, and Figure corrections.

Author Response

Thank you for your comments. We have addressed them point by point in the attached document.

Reviewer 2 Report

Comments and Suggestions for Authors

A long term performance of some PM2.5 sensors was performed in this work and the result was reported in this manuscript. The content is consistent, but it is not logically organized or presented, particularly in Section 3. Overall, the abundance of information provided is appreciable. However, the reviewer suggests reorganizing the content, as some essential information resides in the supplementary material, while certain details in the main text could be omitted. Specific comments of section and subsection of the paper are listed below.

Introduction

It would be useful to add in the introduction that we are talking about low cost sensors and that there is often a lack of information on the factory calibrations that are made. There is also a type on the reference 17, the capital letter is missing on the reference section.

Materials and methods

Study design overview

Could you clarify the number of sensors tested? Was it 54? 

Sensors selected

There is a typo on line 83. Consider moving Table S1 to the main text or adding the columns for 'Measured Pollutants' and 'Sampling Interval' to Table 1. Alternatively, you could move details like the firmware version to the supplementary information.

Long-term monitoring sites selected

This section could benefit from incorporating material presented in the supplementary information, such as Figure S1 and the detailed data for each site. The colocation of sensors in different climate regions is particularly interesting and could be given greater emphasis. Additionally, it would be helpful to reiterate the colocation period, for example, in Figure S2. Line 107 is not clear.

Data processing and analysis

A quality assurance procedure is mentioned; please provide additional details or a reference. Lines 132-133 are unclear and could benefit from clarification. Lastly, Table S26 is important for this work; consider moving it to the main text or rephrasing the paragraph to include more information about the type of flag proposed.

Results & Discussion

Data completeness and common failure points 

In the reviewer's opinion, this entire section could be moved to the supplementary material. At most, the paragraph on lines 197-205 could remain, though it might be more appropriately included in the conclusions/limitations.

Overall performance by site

In the reviewer's opinion, this entire section could be moved after the 'Common Data Issues' section. Additionally, in the caption of Figure 1, it is mentioned that points above 200 µg/m³ were excluded; please specify the reason for this exclusion.

Common Data Issues

This section could benefit from being summarized. Additionally, it would be interesting to give more prominence to the subsection on the influence of RH. Furthermore, the section on variability in bias could be expanded into its own subsection, as the authors themselves highlight that the results vary significantly depending on whether hourly or daily averages are used. This new section could also include a discussion on overall performance as a starting point.

Limitations

This section could be included in conclusions section.

Author Response

Thank you for your comments. We have addressed them point by point in the attached document

Reviewer 3 Report

Comments and Suggestions for Authors

This paper presented a study of PM gas sensors across several states in the US alongside the regulatory monitors. The results have shown both the statistical and technological sides in comparison with the federal methods. Therefore, this work would be of interest to readers in the sensor research field, as well as the regulatory department. I would recommend it for publication after minor corrections.

1) Please do double check some of the format in the manuscript, such as extra space in page 16, centring table, etc.

2) Include some of the sensor device photos or site photos to give a better understanding. There are some in the supplementary materials, but do include one or two in the main content.

3) The plots have lots of information, so either add legends (for different sensors or data type) or more explanations in the captions.

4) The sensors are tested alongside the regulatory method, so beside the performance of the sensor units, it will be better to emphasize a bit more on the comparison study between the two systems.

Author Response

(The authors gave the same response as above.)

Round 2

Reviewer 2 Report

Comments and Suggestions for Authors

The authors provided thorough and detailed responses to all comments.